# Factors Influencing SARS-CoV-2 Vaccine Acceptance and Hesitancy in a Population-Based Sample in Italy

**DOI:** 10.3390/vaccines9060633

**Published:** 2021-06-10

**Authors:** Marco Del Riccio, Sara Boccalini, Lisa Rigon, Massimiliano Alberto Biamonte, Giuseppe Albora, Duccio Giorgetti, Paolo Bonanni, Angela Bechini

**Affiliations:** 1Medical Specialization School in Hygiene and Preventive Medicine, University of Florence, 50134 Florence, Italy; lisa.rigon@unifi.it (L.R.); massimilianoalberto.biamonte@unifi.it (M.A.B.); giuseppe.albora@unifi.it (G.A.); duccio.giorgetti@unifi.it (D.G.); 2Department of Health Sciences, University of Florence, 50134 Florence, Italy; sara.boccalini@unifi.it (S.B.); paolo.bonanni@unifi.it (P.B.); angela.bechini@unifi.it (A.B.)

**Keywords:** SARS-CoV-2, COVID-19, vaccine acceptance, vaccine hesitancy, survey

## Abstract

Vaccination against SARS-CoV-2 represents an effective and safe tool to protect the population against the disease; however, COVID-19 vaccine hesitancy could be a major barrier to achieving herd immunity. Despite the severity of the current pandemic, the population’s intention to get vaccinated against COVID-19 is still not clear. The aim of this study was to evaluate the intention to get vaccinated against COVID-19 among a convenience sample of the general population resident in Italy and the factors associated with hesitancy and acceptance of the vaccine in the context of the current pandemic before the rolling out of COVID-19 vaccines. An anonymous online survey was diffused among a general adult population living in Italy. Participants aged 18 or older and living in Italy were considered eligible. Incomplete questionnaires were excluded. Overall, 7605 valid questionnaires were collected. Most of the participants (81.9%) were inclined to get vaccinated; male sex (OR 1.38, 95% CI 1.12–1.71), a high level of trust in institutions (OR 3.93, 95% CI 2.04–7.83), and personal beliefs about high safety of COVID-19 vaccines (OR 56.33, 95% CI 31.57–105.87) were found to be among the significant predictors of COVID-19 acceptance. These data could help design larger studies to address the problem of COVID-19 vaccine hesitancy in the current pandemic.

## 1. Introduction

Coronavirus disease (COVID-19) is a viral disease caused by SARS-CoV-2 (severe acute respiratory syndrome coronavirus 2), a new strain of coronavirus first identified in humans in January 2020. The WHO declared the coronavirus outbreak a pandemic on 11 March 2020 [1].

Italy was one of the first Western countries to be severely affected by the coronavirus pandemic: at first, the pandemic mainly affected the northern and central parts of the country and then spread nationwide. As of today, more than 1 million people in Italy have been infected with SARS-CoV-2 and at least 117,000 people have died [2], but the spread of COVID-19 could be larger than officially reported [3].

Vaccination against SARS-CoV-2 represents an effective and safe tool to protect the population against the disease: as of 21 February 2021, three COVID-19 vaccines with greater than 60% efficacy to reduce symptomatic infection risk have been approved in the EU [4,5,6] and many other candidate vaccines are in phase 3 trials [7].

However, COVID-19 vaccine hesitancy represents a major barrier to achieving herd immunity; despite the current pandemic, the population’s intention to get vaccinated against COVID-19 is still not clear. Vaccine hesitancy can be defined as a “delay in acceptance or refusal of vaccination despite availability of vaccination services”, with several factors affecting it, according to the Strategic Advisory Group of Experts on Immunization (SAGE) [8,9].

A systematic review of surveys of the general population revealed that the willingness to receive a pandemic vaccine ranged from 8% to 67% [10].

In Italy, vaccine hesitancy is often related to spreading fake news on the disease and related vaccinations [11]. During the first SARS-CoV-2 pandemic, fake news had a significant impact on the Italian population, and they still represent one of the most contributing factors to the skepticism on vaccinations [12].

This is a particular concern in Italy, which has been shown to be a country with a significant presence of vaccine hesitancy [13].

A mass vaccination campaign is currently the only way to boost immunity in order to protect the population from SARS-CoV-2 infection, and high rates of acceptance and coverage are needed. The aim of this pilot study was to evaluate the intention to get vaccinated against COVID-19 among a convenience sample of the general population resident in Italy and the factors associated with hesitancy and acceptance of the vaccine in the context of the current pandemic.

## 2. Materials and Methods

An anonymous online survey was conducted from 11 December to 15 December 2020 among an adult general population. The questionnaire was shared through social platforms (Facebook, Instagram, WhatsApp, Telegram) and using a regional website that aims at enhancing the knowledge and the awareness about vaccination in the general population (https://www.vaccinarsintoscana.org accessed on 18 April 2021).

Participants aged 18 or older and living in Italy were considered eligible. Incomplete questionnaires were excluded. Participation in the study was voluntary, and data were collected anonymously. All participants provided online informed consent to be included in the study.

### 2.1. Questionnaire

In a first section, along with socio-demographic variables, the questionnaire addressed aspects such as trust in institutions, vaccine hesitancy and adherence to the 2020–2021 influenza campaign.

As stated by the WHO SAGE, vaccine hesitancy refers to a complex phenomenon defined as delay in acceptance or refusal of vaccine despite the availability of a vaccination service. According to the WHO’s definition and other studies published on hesitancy, in this study, we evaluated participants’ self-reported vaccine hesitancy using two adapted questions: we asked whether the participants have ever refused or postponed a vaccination recommended by a physician for themselves or a child because they considered it useless or dangerous and whether they would advise others against a vaccination, even if recommended [14,15,16]. Subjects answering “yes” to at least one of this two questions were considered hesitant.

Trust in institutions was assessed with a Likert scale by asking participants how much they trust the institutions (possible answers ranged from “1—I have no trust at all in institutions” to “4—I have full trust in institutions”).

In a second section, we used Likert scales to first assess the intent to be vaccinated against the novel coronavirus (possible answers ranged from “1—surely not” to “5—surely”); then, we evaluated possible determinants of a vaccine hesitancy or acceptance such as the perceived relevance of the vaccination in fighting against the virus and the perceived safety of the vaccination against COVID-19. We also assessed the possibility that the respondent or a family member or acquaintance had contracted the disease or had been hospitalized because of it.

### 2.2. Statistical Analysis

Descriptive statistics were conducted to generate summary tables for study variables.

The item related to the willingness to be vaccinated against COVID-19, previously expressed with five answers, was transformed into a binary variable, grouping answers “1—surely not”, “2—probably not”, and “3—I am not sure” in “no/not sure”, and “4—probably” and “5—surely” in “yes/probably”.

In order to assess the predictors of this new, dichotomous variable, we performed both single and multiple analyses.

In particular, 12 single logistic regression analyses were performed, one for each item of the questionnaire. Variables that were found to be significantly associated with the dependent variable were included in a backward stepwise procedure to perform a multiple logistic regression model: results are presented as an odds ratio (OR) with 95% confidence interval (95% CI).

For all the analyses, a *p*-value < 0.05 was considered significant. Statistical analyses were conducted using RStudio 1.2.5033/RStudio Team, 2019. (RStudio: Integrated Development for R. RStudio, Inc., Boston, MA, USA; URL: http://www.rstudio.com accessed on 18 April 2021).

## 3. Results

Of 7656 collected questionnaires, 0.25% (n = 19) had to be excluded as participants denied consent to the treatment of personal data, while 0.42% (n = 32) were excluded because respondents stated that they do not live in Italy. A total of 7605 questionnaires were considered available for data collection and analysis.

Table 1 shows the summary statistics of the sociodemographic profile of the study participants.

Of the total respondents, 64.9% (n = 4939) were females, while median age was 47.0 years (IQR: 34.0–58.0 years); 35.9% (n = 2729) were healthcare professionals (HCP), and three out of four (n = 5706) were employed when completing the questionnaire. With regard to the level of education, most respondents (39.4%, n = 3000) had a high-school qualification, followed by those with a master’s degree (35.5%, n = 2686), a bachelor’s degree (14.1%, n = 1070), those with a secondary school qualification (7.6%, n = 578), with a PhD (3.1%, n = 239), and finally those with a primary school qualification (0.4%, n = 32). When asked about their trust in institutions, the most selected answer was “3—I have enough trust in institutions” (65.6%, n = 4633), while the mean and the median of the item were 2.8 and 3, respectively.

Table 2 shows the personal beliefs about vaccinations and the willingness to get vaccinated against COVID-19.

Most of the participants declared they had never “refused or postponed a vaccination recommended by a physician for themselves or a child because they considered it as useless or dangerous” (88.0%, n = 6695), while 91.0% (n = 6952) stated that they would not advise others against a vaccination, even if recommended.

Combining these results, as specified in the methods section, we identified 1204 hesitant individuals (15.8% of the sample).

Moreover, less than a half (43.5%, n = 3309) of participants stated that they had been vaccinated against influenza in the 2020–2021 season.

When asked if they will vaccinate against COVID-19, if recommended for them, 6231 (81.9%) subjects answered that they would get vaccinated (4908, 64%, answered “I will surely get vaccinated”, while 1323, 17.4%, answered “I will probably get vaccinated”); 1374 (18.1%) were hesitant about COVID-19 vaccination or even against it.

A slightly higher percentage of subjects willing to be vaccinated against COVID-19 was found among HCPs (82.7% of the HCPs, n = 2256; Table 3).

The exploration of the personal beliefs and personal experiences about COVID-19 showed that 3417 subjects (44.9%) thought that COVID-19 vaccines would be safe (n = 3580, 47.1% declared they “don’t know” and n = 608, 8.0% answered “no”); most of the participants (n = 5818, 76.5%) were sure that the vaccination would be “absolutely relevant” in fighting the COVID-19 pandemic.

Moreover, it emerged that 6546 (86.1%) participants knew someone that had been infected by SARS-CoV-2 (including themselves); almost a half of our sample (n = 3353, 44.1%) also knew someone that had been hospitalized for the same reason (including themselves).

According to multiple logistic regression (Table 3), factors independently found associated with higher COVID-19 vaccine acceptance were of male sex (OR 1.38, 95% CI 1.12–1.71) and had a higher level of trust in institutions (OR 3.93, 95% CI 2.04–7.83); moreover, subjects that got vaccinated against influenza had less chances to be against COVID-19 vaccination (OR 2.09, 95% CI 1.68–2.58).

Employed individuals and hesitant subjects had less chance to be willing to get vaccinated against COVID-19 (OR 0.77, 95% CI 0.61–0.95% and OR 0.13, 95% CI 0.11–0.17). Finally, those who thought that this vaccination would be safe and/or relevant to fight the COVID-19 pandemic were more likely to be willing to be vaccinated.

Knowing someone that had been infected by COVID-19 or hospitalized due to COVID-19 was not significantly associated with increased vaccine acceptance, as well as being HCP, since COVID-19 vaccine acceptance in non-HCPs was almost as high as in HCPs.

## 4. Discussion

Vaccination is surely one of the most relevant public health interventions, but its acceptance varies with space, time, social class, ethnicity, and contextual human behavior [17].

In the current context, COVID-19 vaccine hesitancy may be the most important limiting factor in efforts to control the pandemic and its effects on health [18,19].

Our pilot study used a web-based questionnaire to evaluate the factors influencing SARS-CoV-2 vaccine acceptance and hesitancy in a population-based sample in Italy: there are still few studies exploring the intention to receive the COVID-19 vaccines before their availability and our results are consistent with those of other studies conducted in Italy and similar countries, although a large variability in COVID-19 vaccine acceptance rates have been reported worldwide, varying from 40% up to >90% [20,21,22].

In particular, 64.5% of our sample declared that they would certainly be vaccinated when possible, while 17.4% answered “probably”, outlining a large percentage of the population (81.9%) willing to be vaccinated, a result in line with another study recently published in Italy [23] and with current data regarding those categories that were first involved in the vaccination campaign, such as the elderly category; more than 80% completed the vaccination cycle and more than 90% received at least 1 dose [24].

Identifying factors that promote acceptance can help policymakers boost COVID-19 vaccination, and the study results reported here provide further evidence of the major role that some determinants of health and personal beliefs can play in vaccine acceptance and hesitancy.

Among the most described positive predictors for accepting the vaccination against SARS-CoV-2 there is the self-perception of a high risk of severe COVID-19: HCP, especially those who work with COVID-19 patients, probably perceiving a greater risk of getting infected, are generally more likely to get vaccinated [25,26].

A total of 82.7% of the HCPs that participated in this survey were determined to get vaccinated, and this figure is consistent with current Italian data (to date, 79.4% of HCPs are vaccinated with a full two-dose immunization course, and the percentage is higher if we consider those that only received the first dose) [24].

Even if previous studies have found conflicting results concerning COVID-19 vaccine hesitancy among HCP [27], they were slightly more inclined than non-HCP to get vaccinated in our sample, even if this result was not statistically significant (OR 1.08, 95% CI 0.96–1.22).

Moreover, as reported by other studies, acceptance was higher in males than females (OR 1.38, 95% CI 1.12–1.71). Explanations may vary, including a higher risk perception in males, due to the higher risk of severe COVID-19 described for men [28,29].

Both previous vaccination against influenza and high level of trust in government/institutions have been studied as factors associated with COVID-19 vaccine acceptance, with results that are consistent with our findings [14,30].

In fact, as well as previous influenza vaccination (OR 2.09, 95% CI 1.58–2.68), a high level of trust in institutions resulted in being a strong predictor of vaccine acceptance in our sample (OR 3.93, 95% CI 2.04–7.83).

Similarly, other studies showed how those who did not intend to get vaccinated against COVID-19 had lower level of trust in institutions [31].

For this reason, and when considering the lack of trust in government that has been found by many studies as predictor of a higher level of COVID-19 vaccine hesitancy, it is necessary to pay attention to the fact that part of our sample (almost the 30%) claimed to have no (or low) trust in institutions: higher trust in institutions and public health authorities may enable people to accept the short-term individual costs associated with vaccination to address the solution of common problems in the community [32].

Factors independently found associated with vaccine hesitancy against COVID-19 vaccine in our sample included being hesitant about other vaccinations and being employed; this is not uncommon, and it could be possible that those who are not working consider the vaccine as a factor that could facilitate their return to work [33].

Individuals that took part in recent studies were generally concerned about vaccines’ safety, usually because their fast development. The most frequently remarked safety observations included quality control and potential side effects [22,34]. It is therefore not surprising that, in our sample, the fact of considering the vaccine as safe was a strong predictor of vaccine acceptance (OR 56.33, 95% CI 31.73–105.87); however, it is important to underline how only a small number of the participants considered it unsafe (8.0%) and that most of the participants thought that it would be a key factor in fighting against the disease (76.5%). This is quite surprising, considering that the survey was conducted in a period in which televisions, newspapers, and media were reporting conflicting information regarding the efficacy, safety, and availability of COVID-19 vaccines.

Age, study level, and knowing people infected or hospitalized due to COVID-19 were not predictors of higher or lower vaccine acceptance in our sample.

### Limitations and Strengths

Our pilot study has several limitations; first, the study involved a convenience sample of Italian adults, and it was distributed online, thus being accessible to web-users only. Therefore, the findings may not be representative of the whole population.

Second, it is a cross-sectional study and depicts a picture of the community response at the point of the survey; moreover, almost 1 participant out of 10 (9%) reported being “not sure” about the intention to uptake the COVID-19 vaccination. The real intention could be different when the vaccine is available or when information about specific COVID-19 vaccines change over time. Finally, the survey took place in December 2020, right before any COVID-19 vaccine was available.

Interestingly, the survey was realized before the beginning of the Italian anti-COVID-19 vaccination campaign: on one side it represents the beliefs of our sample before the vaccination started, on 27 December, along with the polarization of the public debate about it. On the other hand, it could potentially not be representative of the current situation after months of debates about COVID-19 vaccine efficacy and safety. As a matter of fact, it only represents the general beliefs about COVID-19 vaccinations right before the beginning of the vaccination campaign, since no specific questions were asked to assess if participants were concerned about a certain type of vaccine or if the opinions towards COVID-19 vaccines have changed during the period of vaccine development.

Despite these limitations, our findings are novel and relevant, and our study is among the first reporting COVID-19 vaccine willingness among a large sample of adults living in Italy and in analyzing the factors that influence COVID-19 vaccine acceptance and hesitancy before the vaccine availability and the dissemination of more in-depth efficacy and safety data.

## 5. Conclusions

The results of this study show high rates of vaccine acceptance among adults living in Italy. In particular, factors such as trust in institutions, previous influenza vaccination, and the beliefs about vaccination safety and efficacy in fighting against the pandemic resulted in being strong predictors of vaccine acceptance.

Large studies, representative of the Italian population, are needed to draw conclusions: if these results will be confirmed and aiming to even increase the number of individuals willing to be vaccinated against COVID-19, it will be important to reinforce some communication characteristics of the current vaccination campaign. It should not only be evidence-based, but it should also clearly and effectively explain to the population the importance of vaccination against COVID-19.

The data outlined in this manuscript can help to design larger studies to better understand the factors that influence COVID-19 vaccine acceptance and to address the problem of COVID-19 vaccine hesitancy in the current pandemic.

## Figures and Tables

**Table 1 vaccines-09-00633-t001:** Sociodemographic characteristics of the study participants.

Characteristics	NA	N or Median	% or IQR
Age	0	47.0	34.0–58.0
*Sex*	43		
Male		2623	34.5
Female		4939	64.9
*Study title*	0		
Primary school		32	0.4
Secondary school		578	7.6
High school		3000	39.4
Bachelor’s degree		1070	14.1
Master’s degree		2686	35.3
PhD		239	3.1
*Healthcare professional*	0		
Yes		2729	35.9
No		4876	64.1
*Trust in institutions*	0		
“4—I have full trust in institutions”		936	13.2
“3—I have enough trust in institutions”		4633	65.6
“2—I have little trust in institutions”		1702	24.1
“1—I have no trust in institutions”		334	4.7
*Employment status*	0		
Currently unemployed/retired		1899	25.0
Currently employed		5706	75.0

**Table 2 vaccines-09-00633-t002:** Personal beliefs about vaccinations and the willingness to vaccinate against COVID-19.

Items	NA	N or Median	% or IQR
*“Have you ever refused or postponed a vaccination recommended by a physician for yourself or a child because you considered it as useless or dangerous?”*	0		
No		6695	88.0
Yes		910	12.0
*“Would you advise others against a vaccination, even if recommended?”*	0		
No		6952	91.4
Yes		653	8.6
*“Did you get vaccinated against influenza in the 2020–2021 season?”*	0		
No		4296	84.2
Yes		3309	15.8
*“Will you vaccinate against COVID-19 when a vaccine is approved, available, and recommended for you?”*	0		
Surely not		345	4.5
Probably not		353	4.6
I am not sure		676	8.9
Probably		1323	17.4
Surely		4908	64.5
*“Have you or others you know been infected by SARS-CoV-2?”*	0		
No		1059	13.9
Yes		6546	86.1
*“Have you or others you know been hospitalized because of COVID-19?”*	0		
No		4252	55.9
Yes		3353	44.1
*“Do you think COVID-19 vaccines will be safe, once approved?”*	0		
No		608	8.0
I don’t know		3580	47.1
Yes		3417	44.9
*“How relevant do you think the COVID-19 vaccination will be in fighting against the pandemic?”*	0		
Not relevant at all		150	2.0
Not very relevant		346	4.5
Quite relevant		1291	17.0
Absolutely relevant		5818	76.5

**Table 3 vaccines-09-00633-t003:** Variables associated with the willingness to be vaccinated against COVID-19. From the left: the first column shows the variables included in the analyses. The second column shows the “willingness to be vaccinated” among the categories reported in each item. The third and the fourth columns show the results of the single logistic regression analyses and the results of the multiple logistic regression analysis.

Variable	Willingness to Be Vaccinated		Single Logistic Regression Analyses				Multiple Logistic Regression Model			
	No/Not Sure (%) Total: 1374 (18.1)	Yes/Probably (%) Total: 6231 (81.9)	OR	Lower 95% CI	Upper 95% CI	*p*-Value	OR	Lower 95% CI	Upper 95% CI	*p*-Value
Age	-		0.995	0.990	0.998	0.01	0.995	0.988	1.001	0.13
Sex *										
Female	959 (19.4)	3980 (80.6)	1.00				1.00			
Male	387 (14.8)	2236 (85.2)	1.39	1.22	1.59	<0.001	1.38	1.12	1.71	0.002
Study										
Elementary school	5 (15.6)	27 (84.4)	1.29	0.54	3.83	0.598				
Secondary school	140 (24.2)	438 (75.8)	0.75	0.61	0.93	0.01				
High school	580 (19.3)	2420 (80.7)	1.00							
Bachelor’s degree	221 (20.7)	849 (79.3)	0.92	0.98	2.05	0.35				
Master’s degree	393 (14.6)	2293 (85.4)	1.40	1.21	1.61	<0.001				
PhD	35 (14.6)	204 (85.4)	1.40	0.78	1.10	0.07				
Healthcare professional										
No	901 (18.5)	3975 (81.5)	1.00							
Yes	473 (17.3)	2256 (82.7)	1.08	0.96	1.22	0.20				
Trust in institutions										
1 (No trust)	243 (72.8)	91 (27.2)	1.00				1.00			
2	648 (38.1)	1054 (61.9)	4.34	3.36	5.66	<0.001	1.69	1.12	2.58	0.014
3	459 (9.9)	4174 (90.1)	24.29	18.80	31.60	<0.001	2.71	1.79	4.11	<0.001
4 (Full trust)	24 (2.6)	912 (97.4)	101.47	64.53	166.28	<0.001	3.93	2.04	7.83	<0.001
Employed										
No	286 (15.1)	1613 (84.9)	1.00				1.00			
Yes	1088 (19.1)	4618 (80.9)	0.75	0.65	0.86	<0.001	0.77	0.61	0.95	0.02
Hesitant subject										
No	185 (3.3)	5430 (96.7)	1.00				1.00			
Yes	1189 (59.7)	801 (40.3)	0.020	0.019	0.027	<0.001	0.13	0.11	0.17	<0.001
Influenza vaccination in 2020–2021										
No	1122 (26.1)	3174 (73.9)	1.00				1.00			
Yes	252 (7.6)	3057 (92.4)	4.28	3.71	4.97	<0.001	2.09	1.68	2.58	<0.001
“Have you or others you know been infected by SARS-CoV-2?”										
No	237 (22.4)	822 (77.6)	1.00							
Yes	1137 (17.4)	5409 (82.6)	1.37	1.17	1.60	<0.001	0.93	0.74	1.23	0.6
“Have you or others you know been hospitalized because of COVID-19?”										
No	634 (18.9)	2719 (81.1)	1.00							
Yes	740 (17.4)	3512 (82.6)	1.11	0.98	1.24	0.09				
“Do you think COVID-19 vaccines will be safe, once approved?”										
No	542 (89.1)	66 (10.9)	1.00				1.00			
I don’t know	816 (22.8)	2746 (77.2)	27.81	21.45	36.64	<0.001	4.35	3.19	6.01	<0.001
Yes	16 (0.5)	3401 (99.5)	1745.58	1034.53	3149.07	<0.001	56.33	31.73	105.87	<0.001
“How relevant do you think the COVID-19 vaccination will be in fighting against the pandemics ?”										
Not relevant at all	145 (96.7)	5 (3.3)	1.00				1.00			
Not very relevant	322 (93.1)	24 (6.9)	2.16	0.87	6.52	0.12	1.30	0.42	5.31	0.68
Quite relevant	541 (41.9)	750 (58.1)	40.20	18.18	113.94	<0.001	8.08	2.81	30.98	<0.001
Absolutely relevant	366 (6.3)	5452 (93.7)	431.99	195.52	1223.75	<0.001	27.99	9.77	107.04	<0.001

* 43 observations deleted due to missingness, as reported in Table 1.

## Data Availability

Data supporting reported results are available upon request to the corresponding author. Data were collected and managed in aggregated form according to European Union Regulation 2016/679 of European Parliament and the Italian Legislative Decree 2018/101.

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
