# Peer review of "Factors Influencing SARS-CoV-2 Vaccine Acceptance and Hesitancy in a Population-Based Sample in Italy"

_vaccines, 2021, doi:10.3390/vaccines9060633_

Round 1

Reviewer 1 Report

This article is a pilot study assessing the awareness and willingness about COVID-19 vaccination among general Italian population. The topic is interesting, however, there are certain limitations that need more explanations. Detailed comments are written below.

Major:

  1. The author said that vaccine hesitancy is related to the fake news on the disease and vaccine (line 49). I wonder why the author did not ask about fake news related COVID-19 vaccination. It might show an interesting result.
  2. What is the actual rate of the COVID-19 vaccination in Italy? How about compare it with the result of this study in discussion (suggestion: in line 178-181).
  3. line 134-135 & 142-143. There is no data about the number of the hesistants and  willingness of HCPs in table 2.
  4. Method and Result of table 3 (multiple regression analysis): I assume that the dependent variable of the logistic regression analysis is 'hesistants', and the independent variables are the variables. Please clarify the statistical method. Did you run the logistic analysis 11 times, or at once with the whole variables? I suggest a multiple logistic regression analysis with adjustment. Especially, age and sex are important factors that influence on vaccination rate, generally, those factors should be taken as adjusted variables. The statistical analysis and table 3 needs more improvement.
  5. line 153-160. The sentences are not follow the sequences of the table and hard to read or understand. 
  6. In table 3, the OR of 101.47 in 4 (full trus) of 'trust in institutions', and 1745.58 in Yes of 'do you think COVID-19 vaccines will be safe?' are noticeable. I think the results might be caused by the small number of those subjects in 'hesistants' group. However, the author did not show the result of the differences of the characteristics between the 'hesistants (N=1,204)' and 'non-hesistants'.
  7. line 186-197. As the author described in the result (line 161) and table 3, being HCP was not associated with increased vaccine acceptance. However, the real vaccination rate among HCPs are high in Italy. More explanations are needed for this gap.
  8. line 198~. How about the difference of vaccination rate between male and female for other vaccines (such as influenza)? Several studies showed opposite results. (ex. Journal of Primary Care & Community Health Volume 11: 1–6. )
  9. limitation: as the authors described in discussion, this study has a selection bias caused by the web-based recruitment. The mean age of the subjects was 47y, which is too young to generalize. Older people are not used to using the internet.

Minor:

  1. I wonder the reason why only 15.8% of the subjects got vaccinated against influenza in the 2020-2021 season. Is the rate of Influenza vaccination in the whole Italian population similar?
  2. Too much paragraphing. It makes hard to understand the context.
  3. line 199. typo: 95% IC -> 95% CI

Reviewer 2 Report

This paper describes a highly topical issue, that of Covid-19 vaccine hesitancy.  The data were collected before the vaccine campaign started in Italy therefore provides a useful benchmark for comparison to later views which could change the acceptance rate in either direction - hesitancy could increase because of concern over side effects, or decrease once the evidence of side effects being rare and the effectiveness in disease reduction emerges.

One very important anomaly.  In Table 1 it gives employment status as 25% employed, 75% unemployed/ retired, but the text in line 118 says the opposite.  I'm assuming the data in the table is the wrong way round because otherwise it would not fit with numbers of HCP and therefore working on this assumption the other data analyses fit into place.

Minor points.  Line 29 should read 'declared'. 

Line 67 'participation in the study'? rather than 'adhesion to'?

Line 74 - as the abbreviation WHO SAGE has already been introduced in line 45 don't need it again in full.

Table 3 typo 'institutions' 

Ref 8 - looks like duplicated text, but I think there is a full stop missing between the two Vaccine Hesitancys

Ref 14.  Should be full stop not line after WHO

Reviewer 3 Report

The manuscript titled “Factors influencing SARS-CoV-2 vaccine acceptance and hesitancy in a population-based sample in Italy” evaluates the intention to get vaccinated against COVID-19 among general population in Italy and factors associated with hesitancy and acceptance. This is an important field of investigation as this pandemic has affected all the countries and vaccination against SARS-CoV-2 represents an effective and safe tool to protect the population against the disease moving forward. The data outlined in this manuscript can help to design larger studies to address the problem of COVID-19 vaccine hesitancy in the current pandemic. The manuscript is a great addition to the current field and has been written well. However, the survey questionnaires do not highlight the main response for vaccine hesitancy.

Recommendation: Accept after major revisions

Major Comments:

  • The survey questions do not highlight the major reason of vaccine hesitancy in the overall population. The survey questions should have included aspects related to “Timelines associated with vaccine development” or is the hesitancy related to a particular vaccine as there have been mRNA as well as Viral vector-based vaccines.
  • The authors need to break down the hesitancy level in participants highlighting if they are concerned about a certain type of vaccine in the manuscript.

Minor Comments:

  • There are some spelling mistakes throughout the manuscript. For example, “Trust in istitutions” need to be changed to Trust in Institutions.
  • The authors should have included questions in the survey that would have helped them understand what initiatives can help to remove vaccine hesitancy among people.

Round 2

Reviewer 1 Report

The authors revised the manuscript according to the reviewer's comments.

Table 3 has been improved, however, the 'ref' in the first column should be removed. Because there are '1.00' as the reference values in the table 3. 

Thank you.

Author Response

Dear reviewer, 

thank you for appreciating our work. 

Following your suggestion, table 3 has been modified. 

Reviewer 3 Report

All my previous contingencies have been addressed.

Recommendation: Accept

Author Response

Dear reviewer, 

thank you for appreciating our work.